# Functions of Enyolreductase (*ER*) Domains of PKS Cluster in Lipid Synthesis and Enhancement of PUFAs Accumulation in *Schizochytrium limacinum* SR21 Using Triclosan as a Regulator of *ER*

**DOI:** 10.3390/microorganisms8020300

**Published:** 2020-02-21

**Authors:** Xueping Ling, Hao Zhou, Qinghua Yang, Shengyang Yu, Jun Li, Zhipeng Li, Ning He, Cuixue Chen, Yinghua Lu

**Affiliations:** 1Department of Chemical and Biochemical Engineering, College of Chemistry and Chemical Engineering, Xiamen University, Xiamen 361000, China; 18014936521@163.com (H.Z.); 18859256912@163.com (Q.Y.); yushengyang218@163.com (S.Y.); lijun0308@163.com (J.L.); hening@xmu.edu.cn (N.H.); cxchen@xmu.edu.cn (C.C.); 2The Key Lab for Synthetic Biotechnology of Xiamen City, Xiamen University, Xiamen 361000, China; 3College of Food and Biological Engineering, Jimei University, Xiamen 361000, China; lizhipeng39@gmail.com; 4Fujian Collaborative Innovation Center for Exploitation and Utilization of Marine Biological Resources, Xiamen 361000, China

**Keywords:** Enyolreductase (*ER*), polyunsaturated fatty acid, *S. limacinum*, triclosan, metabolomics

## Abstract

The polyketide synthase (PKS) cluster genes are supposed to synthesize polyunsaturated fatty acids (PUFAs) in *S. limacinum.* In this study, two enyolreductase (*ER*) genes located on PKS cluster were knocked out through homologous recombination to explore their functions. The knock-out of *OrfB-ER* (located on *OrfB* subunit) decreased lipid content and had obvious decrease on PUFAs content, indicating *OrfB-ER* domain played a vital role on PUFAs synthesis; the knock-out of *OrfC-ER* (located on *OrfC* subunit) decreased SFAs content and increased total lipid content, indicating *OrfC-ER* domain was likely to be related with SFAs synthesis, and lipid production could be improved by down-regulating *OrfC-ER* domain expression. Therefore, the addition of triclosan as a reported regulator of *ER* domain induced the increase of PUFAs production by 51.74% and lipids yield by 47.63%. Metabolic analysis indicated triclosan played its role through inhibiting the expression of *OrfC-ER* to reduce the feedback inhibition of SFAs and further to enhance NADPH synthesis for lipid production, and by weakening mevalonate pathway and tricarboxylic acid (TCA) cycle to shift precursors for lipid and PUFAs synthesis. This research illuminates functions of two *ER* domains in *S. limacinum* and provides a potential targets for improving lipid production.

## 1. Introduction

The polyunsaturated fatty acids (PUFAs), mainly including docosahexaenoic acid (DHA; C22:6) and eicosapentaenoic acid (EPA; C20:5), are beneficial to human health. For example, PUFAs may be beneficial in prevention of cardiovascular disease and treatment of mild Alzheimer’s disease [1,2]. Moreover, even though PUFAs treatment may cause mild indigestion and skin disorders, PUFAs are generally safe and well tolerated [3]. In recent years, *S. limacinum* has been widely studied due to its ability to produce significant amounts of total lipids rich in PUFAs. It was reported that *S. limacinum* was able to achieve 40%~70% of the lipid content and 20%~50% of the DHA production [4,5]. Even though other microalgaes such as *Phaeodactylum tricornutum* and *Nannochloropsis oceanica* could reach relatively high lipid content under nitrogen limitation [6], phosphate limitation [7], high salinity [8], and genetic engineering [9], lipid production and PUFAs production were relatively low due to their low biomass. In addition, the high lipid and PUFAs content of *S. limacinum* are beneficial for the novel extraction and purification methods such as the supercritical fluid extraction technique [10]. The heterotrophic microalga *S. limacinum* have been approved to commercially produce DHA mainly used for pharmaceutical products and infant formula, which represent an industrially important source of PUFAs [11]. 

The primary biosynthetic processes responsible for the formation of PUFAs in *S. limacinum* have been identified in two biochemical pathways: The aerobic fatty acid synthesis (FAS) pathway and the anaerobic polyketide synthase (PKS) pathway. The FAS pathway comprises a series of desaturation and elongation steps, which require molecular oxygen, while the putative PKS pathway involves several rounds of reduction, dehydration, reduction, and condensation, which does not require molecular oxygen [12,13]. Some studies proposed that the PKS pathway mainly produce PUFAs in some marine microalgae [5,14]. Like similar enzymes identified from other marine microbes such as *Shewanella*, the PUFAs synthase from PKS pathway in *S. limacinum* is comprised of three subunits (*Orf A-C*), each with multiple catalytic domains (Figure 1A) [12,15,16]. In recent years, there were many studies about the function of these catalytic domains from PKS pathway. For example, Ren et al. found that the lower level of DHA content was observed by deleting acyltransferase (*AT*) domains in *Schizochytrium* sp. HX-308, which indicated that *AT* domains played a key role in DHA synthesis [17]. In addition, some catalytic domains from PKS pathway such as ketoacylsynthase (*KS*) domains and dehydratase (*DH*) domains were comprised of two smaller domains (Figure 1A) [18]. Xie et al. found that the *DH-A* domain (located on *OrfA* subunit) was similar to *DH* domains from PUFAs synthases, while the *DH1-C* and *DH2-C* domains (located on *OrfC* subunit) were more comparable to *E.coli* FabA as a SFAs synthases protein [19]. Similarly, the enyolreductase (*ER*) domains in *S. limacinum* were also comprised of *OrfB-ER* (located on *OrfB* subunit) and *OrfC-ER* domain (located on *OrfC* subunit) (Figure 1A) [20] whose roles for lipid synthesis in *S. limacinum* are completely unknown.

The ER (FabI) in *E.coli* are responsible for the reduction of the double bond in the trans-enoyl-ACP for fatty acid synthesis [21,22]. Metz et al. showed that the *ER* domains of *Schizochytrium* sp. are homologous to the *ER* domain of *Shewanella* SCRC2738, and the protein sequence identity was more than 50% [12]. In *Shewanella*, it was proposed that the *ER* domain was similar to enoylreductase from *Streptococcus pneumoniae*, and could catalyze reduction of the double bond to synthesize EPA by PKS pathway [12,23]. Therefore, the *ER* domains in *Schizochytrium* sp. were considered to play potential roles in PUFA synthesis. In addition, triclosan was widely used to regulate fatty acid synthesis as a reported ER inhibitor [24,25,26]. Escalada et al. and Liu et al. found butenoic acid production was improved by inhibiting the transformation of butenoyl-ACP to butyryl-ACP through adding triclosan [27,28]. Moreover, the addition of triclosan could cause a significant increase of DHA content and lipid content in *Aurantiochytrium* sp. and was also used to increase arachidonic acid (ARA) production in *Mortierella alpina* [29,30]. However, the mechanism of triclosan on fatty acids production has never been reported.

In this study, we explored the functions of the *OrfB-ER* and *OrfC-ER* domains on *S. limacinum* by gene knockout. Based on the roles of *ER* domains, we attempted to add triclosan to regulate PUFAs production, and to clarify the regulatory mechanism of triclosan on fatty acids synthesis using the GC-MS and a multivariate analysis. The study could provide a better understanding of the *ER* action in PUFAs synthesis and a new direction to regulate lipid and PUFAs synthesis on *S. limacinum*.

## 2. Materials and Methods

### 2.1. Strain and Culture Conditions 

*Schizochytrium limacinum* SR21 was purchased from the American Type Culture Collection (Manassas, USA) and maintained on seed broth agar plates. The seed medium (per liter, pH 6.5) contained glucose, 20 g; Na_2_SO_4_, 12 g; yeast extract, 10 g; MgSO_4_, 2 g; (NH_4_)_2_SO_4_, 1 g; KH_2_PO_4_, 1 g; K_2_SO_4_, 0.65 g; KCl, 0.5 g; CaCl_2_·2H_2_O, 0.17 g. The fermentation medium (per liter, pH 6.5) contained glucose, 90 g; Na_2_SO_4_, 12 g; tryptone (OXOID, Massachusetts, USA), 5 g; corn steep powder (Sigma, San Francisco, USA), 5 g; MgSO_4_, 2 g; (NH_4_)_2_SO_4_, 1 g; KH_2_PO_4_, 1 g; K_2_SO_4_, 0.65 g; KCl, 0.5 g; CaCl_2_·2H_2_O, 0.17 g [31]. The stored cells were cultured at 28 °C, 200 rpm for 36 h in the seed medium. After three generations of cultivation, the seed medium (4%, v/v) was transferred to fermentation medium and incubated at 28 °C, 200 rpm, for 120 h or more.

### 2.2. Vector Construction, Transformation and Screening 

The vectors used for disruption of *OrfB-ER* and *OrfC-ER* gene were constructed as shown in Figure 1B,C, and the primers used are listed in Appendix A. The *OrfB* and *OrfC* sequence are listed in Appendix A. First, the upstream and downstream fragments of *OrfB-ER* gene were amplified using B-UF (A)/B-UR (B) and B-DF (C)/B-DR (D), then cloned into the pMD19T Easy Vector. The generated pMD19T-BU and pMD19T-BD were cut using *Sal*I/*Hind*III and *Pst*I/*BamH*I restriction enzyme, respectively. The pBlueZEO vector contained zeocin expression cassette (TEF1 promoter, zeocin gene, CYC1 terminator) was digested with the same restriction enzymes. Next, fragments were ligated at 16 °C overnight using T4 ligase enzyme, generating the targeting vector pBlueZEO-B. The pBlueZEO-C vector was constructed using the same method with the restriction enzymes of *Kpn*I/*Cla*I and *Pst*I/*BamH*I. 

The constructed two vectors were introduced into *S. limacinum* by electrotransformation, which was previously reported by Hong et al. [32]. Some processes were modified as follows: The vectors used for disruption of *OrfB-ER* and *OrfC-ER* gene were linearized with *Sal*I/*BamH*I and *Kpn*I/*BamH*I, respectively. Then, 100 μL competent cells and 3 μg linearized vector were mixed in 0.2 cm electric shock cup, and were shocked 6 ms at 2 kV. After electroporation, the cells were cultivated for 2 to 3 h at 28 °C in seed medium contained 1 M sorbitol. The recovered cells were coated for 3 to 5 days in solid medium containing 30 μg/mL zeocin. Finally, the disrupted strains grown on the zeocin plate were cultured at 28 °C, 200 rpm, for PCR validation and fermentation experiments.

### 2.3. Biomass Determination, Lipid Extraction and Fatty Acid Analysis 

Biomass was determined by gravimetric method. Lipid extraction method was same as Li et al. [5]. Fatty acid methyl esters (FAMEs) were prepared regarding to previous methods [33] with some modifications. First, 5 mL 0.5 M KOH–methanol was added to a tube containing total lipids. The tubes were heated in a water bath at 65 °C for 10 min and then 5 mL 30% BF_3_-ether was added. The tubes were further heated in a water bath at 65 °C for 30 min and then 5 mL hexane was added when the tubes cooled down to room temperature. Then, 16 g/L methyl heptadecanoic acid (Sigma, San Francisco, USA) served as an internal standard and was added to mixtures. Then, the mixtures settled for separation of two phases after adding 1 mL saturated sodium chloride solution for preventing emulsification. FAMEs were applied to a gas chromatograph (Agilent GC 7890, Sacramento, USA) equipped with a 100 m × 0.25 mm capillary column (SP-2560, Sacramento, USA). The column was increased from 140 to 240 °C at 3 °C/min and then maintained at 240 °C for further 30 min. The temperature of the injector and detector were both set at 260 °C. Nitrogen was used as the carrier gas at 20 cm/s. Peaks were identified using authentic standards of supelco 37 component FAME mix (Sigma, San Francisco, USA). Fatty acids were quantified from the peak areas relative to the peak of the internal standard. 

### 2.4. Quantitative Real-Time PCR Analysis (RT-qPCR) 

Total RNA was extracted from 1 mL of *S. limacinum* cells using MiniBEST Universal RNA Extraction Kit (TaKaRa, Osaka, Japan) at 48 and 72 h. cDNA was prepared using EasyScript First-Strand cDNA Synthesis SuperMix (Transgen Biotech, Beijing, China) according to the manufacturer’s protocol. PCR was performed using a TransStart Top Green QPCR SuperMix (Transgen Biotech, Beijing, China). The obtained cDNA was diluted to 100 ng/μL for further real-time PCR analysis. A StepOne Real-time PCR System (Applied Biosystems, Beijing, USA) was used to detect expression of the genes. The RT-qPCR refers to method of Cui et al. [29]. The primers used for RT-qPCR are listed in Appendix A. Reactions without a cDNA template were used as negative controls, and the samples of the wild type strain were used as relative references in calculations.

### 2.5. Enzyme Activities Analysis

The cells were harvested by centrifugation at 10,000× g for 2 min. The pellet was washed by sterile water twice, then grounded with liquid nitrogen. The disrupted cells were centrifuged at 12,000× g for 10 min, and the supernatant was used as enzyme assay. The activity of acetyl-CoA carboxylase (ACC) enzyme was determined using the ACC assay kit (Solarbio, Beijing, China) according to the instructions. ACC could catalyze the formation of malonyl-CoA by acetyl-CoA, NaHCO_3_ and ATP. The activity of ACC was determined by measuring the increase of inorganic phosphorus. The activity of NADP+-dependent glucose-6-phosphate dehydrogenase (G6PDH) was assayed as described by Langdon [34] using continuous spectrophotometric assays following the increase of NADPH at 340 nm. 

### 2.6. GC-MS Analysis

Before GC-MS analysis, sample derivatization was performed according to the method [35] with some modifications. The sample was quickly mixed with cold methanol (−40 °C) to capture the metabolism of cells instantaneously and then centrifuged at 8000× g and 4 °C for 10 min. The cell pellets were washed twice with physiological saline and then stored at −80 °C. The samples were ground to a fine powder under liquid nitrogen. Then, 0.5 mL of cold methanol (−40 °C) was added to 0.1 g of cell powder and mixed thoroughly for 30 s. The mixture was centrifuged at 8000× g for 5 min at 4 °C. The supernatant was resuspended with 0.5 mL of methanol and the mixture was centrifuged at 8000× g for 5 min at 4 °C. The supernatant was stored at −80 °C until use.

The obtained samples and 2.5 μL of internal standard (methyl heptadecanoic acid in n-hexane, 16 mg/mL) were mixed and dried in a vacuum freezer dryer. Methoxyamine hydrochloride (50 μL) in pyridine (50 g/L) was added to the dried sample and incubated at 37 °C for 2 h. Next, the sample was also incubated for 2 h at 37 °C by adding 50 μL of N-methyl-N-(trimethylsilyl) trifluoroacetamide (Sigma, Burlington, USA) and vortex mixing for 30 s.

The sample was analyzed by GC-MS as previously described [36] using an Agilent 7890-5975C GC-MS solution system (Agilent GC-MS 7890-5975C, Sacramento, USA) with a DB-5 capillary column (30 m × 0.25 mm, 0.25 μm film thickness); 1 μL of the sample was injected into the DB-5MS capillary column coated with 5% phenyl and 95% methylpolysiloxane in splitless mode. The column temperature was held at 70 °C for a 2 min delay and then increased to 290 °C at the rate of 8 °C/min, then held for 3 min. Helium was used as carrier gas, and the flow was constant at 1 mL/min. The transfer line and ion source temperatures were 280 and 250 °C, respectively. The mass scan range was 50–600 *m/z*.

### 2.7. Statistical Analysis

The statistical significance between different strains or different groups was presented by T-test. *p* < 0.05 was considered statistically significant. Each experiment was conducted three times. Values are expressed as means ± SD (the standard deviation).

## 3. Results and Discussion

### 3.1. Construction of Deletion Mutants 

The TEF1 promoter and the CYC1 terminator were used to construct zeocin open reading frame (ORF) in plasmids (Figure 1B,C). The linearized plasmids for disruption were transformed to *S. limacinum* by electroporation. The zeocin gene was used to serve as a screening marker, and *S. limacinum* were cultured in seed medium containing 50 μg/mL zeocin. For *OrfB-ER* or *OrfC-ER* mutant strain, nearly 20 transformants were found on plates containing 50 μg/mL zeocin. Fragments of 649 bp corresponding to the partial zeocin resistance cassette were successfully detected in disruption mutants, while not detected in wild type strain (Figure 2A,B). In addition, the results of PCR (*OrfB-ER* 1997 bp, *OrfC-ER* 1972 bp) were detected in the corresponding disruption strain, respectively (Figure 2E), whereas the *OrfB-ER* (1334 bp) and *OrfC-ER* (911 bp) were detected in the wild type strain (Figure 2C,D). It indicated that two *ER* domains on *S. limacinum* were successfully knocked out, respectively. The design of pair of primers PCR is listed in Appendix A.

Moreover, both of the transcription level of *OrfB-ER* and *OrfC-ER* genes were significantly reduced (Appendix A). The reason that two *ER* domains transcriptional activity exists in mutants might be due to the insertional inactivation deletion method (Figure 1B,C). Our results are similar to Ren et al. study in which zeocin gene was inserted in the middle of *OrfB* subunit and the front of the *OrfB* subunit still existed, producing a deformity OrfB enzyme [17]. It further indicated that two *ER* domains on *S. limacinum* were successfully knocked out, respectively.

### 3.2. Biomass and Lipids Accumulation Analysis of Deletion Mutants

Both biomass of *OrfB-ER* and *OrfC-ER* disruption strains were 19.0% and 28.3% lower than wild type strain, respectively (Figure 3A). A similar phenomenon of a significant decrease in biomass was also reported in the knockout of *DH*, *CLF*, and *AT* genes in *S. limacinum* [17,29], which was mainly due to the gene disruption has an impact on lipid synthesis which affects the fluidity of cell membranes. 

As shown in Figure 3B, the lipid content of *OrfB-ER* disruption strain was not only lower than the wild-type strain, but also lower than *OrfC-ER* disruption strain during the whole fermentation process. Moreover, the lipid content of *OrfC-ER* disruption strains surpassed the wild-type strain from 72 h until the end of fermentation, and the obtained maximum of 58.72% was increased by 9.6% (*p* < 0.05) at 72 h comparing to the wild-type strain. These results indicated that the knock-out of *OrfB-ER* gene not only had an suppress on biomass, but also have more reduction on lipid production. On the contrary, even though *OrfC-ER* gene disruption had the similar inhibition on biomass with *OrfB-ER* gene disruption, it caused a small increase in lipid productivity compared to the wild-type strain. Therefore, it is suggested that *OrfB-ER* is likely to play a more vital role in lipid accumulation than *OrfC-ER*. More importantly, it showed that it is possible to increase lipid production by inhibiting *OrfC-ER* gene expression, which guided us to add regulators such as triclosan to regulate the activity of OrfC-ER.

### 3.3. Lipid Profiles Analysis of Deletion Mutants

Due to the lipid accumulation that mainly occurred between 48 h and 72 h [37], and reached the highest lipid content at 72 h (Figure 3B), cells of 48 h and 72 h were selected to analyze fatty acid compositions. As shown in Figure 3C, the disruption of *OrfB-ER* and *OrfC-ER* gene had a significant and different effect on fatty acid compositions. For *OrfB-ER* gene disruption, compared to the control, the content of PUFAs showed the 65.14% (*p* < 0.01) and 48.76% (*p* < 0.01) decrease at 48 h and 72 h. Differently, for *OrfC-ER* gene disruption, SFAs content were always lower than the control, while the proportion of PUFAs had little decrease at 48 h and 72 h. These results illustrated that *OrfB-ER* gene plays an important role in PUFAs biosynthesis while *OrfC-ER* is considered to be mainly related with SFAs synthesis in *S. limacinum*. Interestingly, the content of monounsaturated fatty acids (MUFAs) was significantly increased in both of *ER* genes disruption strains, which was likely to be related with cells membrane fluidity [38]. 

In addition, as shown in Appendix A, the transcription level of *DH* (located on *OrfC* subunit) gene increased obviously in *OrfB-ER* gene disruption strain, while decreased in *OrfC-ER* gene disruption strain. As reported, *DH* domain was beta-hydroxyacyl-acyl carrier protein dehydratases in fatty acid biosynthesis [39], and was more relevant to *FabA* used in SFAs synthesis [12,19]. It further indicated *OrfB-ER* gene was mainly responsible for PUFAs synthesis, whose knock-out shifted the substrate to SFAs pathway due to the competitive relationship between them [5]. Combined with the great decrease of *DH* gene with the fatty acids profiles in *OrfC-ER* mutant strain, it is possible to improve PUFAs synthesis by inhibiting *OrfC-ER* gene expression.

### 3.4. Effects of Triclosan as an Regulator of Enoylreductase (ER) on Growth and Lipid Production

It was found that the addition of 1 μM triclosan at 0 h had little influence on lipid accumulation and biomass, but the addition of 3 μM triclosan in the medium caused an obvious decrease in the growth of *S. limacinum* (Appendix A). When the concentration of triclosan was increased to 5 μM or higher concentration, *S. limacinum* rarely grew, indicating that high concentration of triclosan (≥5 μM) was lethal to *S. limacinum* (Appendix A).

In addition, we analyzed the fatty acid composition to investigate how triclosan affects biomass. Under 3 μM triclosan treatment, the content of C16:0 and C14:0 was reduced markedly, while the content of C14:1 was increased (Appendix A). It indicated that SFAs synthesis was inhibited under triclosan stress, while MUFAs content was increased to improve the cell membrane fluidity to maintain cell growth. Our results were in accordance with previous studies in which high concentration of triclosan had a lethal effect on cells, mainly due to the addition of triclosan at the beginning of fermentation, which had a negative effect on SFAs synthesis involved in the function of cell membrane and further inhibited cell growth [23,30,40].

In order to reduce the inhibition of triclosan on biomass, triclosan was added at 24 h and 48 h. It was observed the addition of triclosan at 24 h had no remarkable effect on biomass (Figure 4A), while it caused an increase of lipid accumulation, especially for 8 μM triclosan treatment, where the total lipids obviously increased by 47.63% (*p* < 0.05) (Figure 4B). As shown in Table 1, for the treatment group, the yield of the total PUFAs, DHA, and EPA presented increases of 51.74% (*p* < 0.01), 49.33% (*p* < 0.05), and 53.38% (*p* < 0.05), respectively. Regarding the addition of triclosan at 48 h, it had little influence on biomass and total lipids production (Figure 4C,D). These results indicated that the optimal addition of triclosan was 8 μM at 24 h, which not only had no effect on biomass, but also caused an increase of total lipid. Therefore, the optimal addition of triclosan was used in follow-up research.

### 3.5. Effects of Triclosan on Fatty Acid Composition

As shown in Table 2, the proportion of SFAs such as C14:0, C16:0, and C18:0 were significantly decreased under 8 μM triclosan treatment, while the content of PUFAs including DHA, EPA, and DPA were obviously increased. As reported, PUFAs are more synthesized by PKS pathway [41,42], and SFAs are more synthesized by FAS pathway [43], both of which require competition for acetyl-CoA as a substrate, ATP as energy, and NADPH as the reducing power [44]. C14:1 under 8 μM triclosan treatment increased over the control, which might be related to cell membrane fluidity under triclosan stress [38]. Moreover, the ratio of PUFAs/SFAs was increased from 64.54% to 76.20% (*p* < 0.01) at 48 h, and a similar change was discovered at 72 h. These results implied that triclosan channeled metabolic flux flow to the PUFAs synthesis. Hence, it indicated that under triclosan stress *S. limacinum* was more beneficial to the synthesis of PUFAs.

### 3.6. Effect of Triclosan on Expression Levels of Related Genes

As shown in Figure 5A, the transcription level of *OrfC-ER* gene showed a 1.55-fold decrease (*p* < 0.05) at 48 h. It indicated triclosan was more likely to inhibit OrfC-ER enzyme activity. The lower transcription level of *OrfC-ER* gene was related to the lower content of SFAs (Table 2). Accordingly, the *FAS* gene, related to SFAs synthesis, was down-regulated 1.23-fold (*p* < 0.05) at 72 h (Figure 5B), which indicated that triclosan could weaken SFAs biosynthesis by suppressing the expression of *OrfC-ER* gene. These results were in accordance with the results of *OrfC-ER* gene disruption in which *OrfC-ER* domain gene was more associated with synthetic SFAs. Interestingly, the transcription level of *OrfB-ER* gene was up-regulated by 2.50-fold (*p* < 0.01) and 1.55-fold (*p* < 0.05) at 48 h and 72 h, which was in accordance with the higher PUFAs production in the triclosan-treatment group. These results indicate that triclosan weakened SFAs synthesis to enhance PUFAs synthesis.

### 3.7. Effects of Triclosan on Key Enzyme Activities

Acetyl-CoA carboxylase (ACC) catalyzes the first committed step towards lipid biosynthesis, converting acetyl-CoA into malonyl-CoA, which played a critical role in fatty acid synthesis. It was reported that ACC enzyme activity could be inhibited by feedback from high palmitoyl-CoA, so a decrease of palmitoyl-CoA draws the fatty acid pathway forward by relieving its feedback inhibition on ACC enzyme activity [45]. In *S. limacinum*, under triclosan treatment, ACC enzyme activity showed an increase of 85.8% (*p* < 0.05) and 69.4% (*p* < 0.05) at 48 h and 72 h, respectively, as compared to the control group (Figure 6A). It meant the improvement of ACC enzyme activity was induced by the addition of triclosan to reduce palmitoyl-CoA synthesis (Table 2), leading to an increase on the accumulation of the de novo synthesized PUFAs. Our results were in accordance with previous studies in which over-expression of stearoyl-CoA desaturase (SCD) alleviated the palmitoyl-CoA inhibition on ACC enzyme activity, thereby increasing lipid accumulation [46].

Furthermore, the high palmitoyl-CoA had a negative influence on pentose phosphate pathway. It is well-known that glucose-6-phosphate dehydrogenase (G6PDH) is a rate-limiting enzyme in pentose phosphate pathway, which provides major reducing power of NADPH for fatty acid synthesis [47,48]. Specifically, some studies found that the pentose phosphate pathway was likely to couple with PKS pathway to supply NADPH for PUFAs synthesis [29]. It was observed that G6PDH enzyme activity was increased by 42.9% (*p* < 0.05) and 46.7% (*p* < 0.05) relative to the control cultures at 48 h and 72 h, respectively (Figure 6B). It indicated that triclosan accreted pentose phosphate pathway by reducing the palmitoyl-CoA accumulation to improve NADPH synthesis, which was beneficial for PUFAs accumulation.

### 3.8. Metabolite Analysis with Triclosan Treatment

GC-MS detected more than 90 putative intracellular metabolites, 37 of which were identified and quantified in all samples at 48 h and 72 h, including intermediate metabolites of tricarboxylic acid (TCA) cycle and mevalonate pathway, amino acids, fatty acids, sugar derivatives, and so on.

The heat maps illustrated the concentrations of these differential metabolites in the 8 μM triclosan treatment and control condition using a log ratio (Figure 7A). It showed PUFAs content such as DHA were significantly promoted, while SFAs content such as myristic acid (C14:0), palmitic acid (C16:0), and stearic acid (C18:0) were reduced. Moreover, glyceryl monostearate and 2-palmitoylglycerol also showed the decrease. Meanwhile, as shown in Figure 7B, it is well-known that glucose, as the most common carbon source of cells, plays a crucial role in carbon storage, biosynthesis, energy supply, and carbon-skeleton construction [49]. In our study, the intracellular glucose content was significantly lower than the control condition at 72 h, indicating that the consumption of glucose was promoted by triclosan. Moreover, the accumulation of fructose as a derivative of glucose under triclosan treatment was greater than triclosan-free treatment, while the content of mannitol, which was also a derivative of glucose, decreased. Mannitol severed as a signaling molecule for oxidative stress and osmoprotectant in plant and fungal physiology, which prevented thiol-regulated enzymes from hydroxyl radical-induced inactivation and maintained redox homeostasis [50,51,52,53]. Meanwhile, Myo-inositol was found to be capable of the carbohydrates storage and to maintain a balance of the intracellular reactive oxygen species [54,55]. Therefore, the lower myo-inositol content was connected with carbohydrate shortage. These results indicated that triclosan could enhance the utilization of intracellular glucose and regulate the intracellular redox balance in fermentation process by changing sugar metabolism.

As the active components in the mevalonate biosynthesis pathway, the significant reduction of squalene meant the mevalonate pathway was weakened by triclosan treatment [56]. There was a competitive relationship between the mevalonate pathway and the fatty acid pathway [57] for utilizing acetyl-CoA as a precursor to synthesize sterol and fatty acid, respectively. When the lipid accumulation was improved, the mevalonate pathway was weakened. Pyruvate was not only a precursor for acetyl-CoA, but also for other metabolites such as lactic acid, acetic acid, and oxaloacetate. Adding triclosan was capable of suppressing lactic acid pathway and promoting the synthesis of acetyl-CoA from pyruvate.

Acetyl-CoA was a crucial intermediate metabolite for some important metabolic pathways, including TCA cycle, fatty acid synthesis pathway, and mevalonate pathway. The addition of triclosan caused the decrease of citrate, succinate, and oxaloacetate, which were some intermediate metabolites of TCA cycle. Moreover, some amino acids, such as oxoproline and arginine, were reduced, which were synthesized by further catalysis of α-ketoglutarate from TCA cycle. These results suggested that triclosan treatment promoted the metabolic flux from TCA cycle to the fatty acids biosynthesis.

Serine and tyrosine were converted from glyceraldehyde-3-phosphate and phosphoenolpyruvic acid, respectively. The decrease of them meant that triclosan treatment could reduce the shunt of some intermediate metabolites including glyceraldehyde-3-phosphate and phosphoenolpyruvic acid in glycolysis pathway and promoted the glycolytic pathway to synthesize pyruvate. Furthermore, the content of proline and 4-aminobutyric acid decreased during the lipid accumulation phase. It was previously reported that proline was able to activate partial anti-stress mechanisms in cells such as oxidative stress and water stress [58,59]. The broadly researched 4-aminobutyric acid have an effect on the supply of NADH, thereby maintaining the intracellular redox balance in the cell [60]. In short, the addition of triclosan enhanced the utilization of intracellular glucose to strength the glycolytic pathway, resulting in the metabolic flux in a direction favorable for fatty acid synthesis.

## 4. Conclusions

This study has demonstrated that *OrfB-ER* and *OrfC-ER* domains on the PKS gene cluster of *S. limacinum* have different functions for fatty acid synthesis by gene knockout. As shown in Figure 8, *OrfB-ER* is considered to play a vital role on PUFAs synthesis while *OrfC-ER* is more relevant to SFAs synthesis, which is potentially utilized to enhance PUFAs production. Thus, the significant increase of lipid production and the proportion of PUFAs to SFAs was achieved by adding triclosan to down-regulate *OrfC-ER* expression, which redirected metabolic flux from MVA pathway and TCA pathway to fatty acid synthesis. The addition of triclosan increased the precursors and NADPH, reducing the power of fatty acid synthesis by increasing ACC and G6PDH enzyme activities, which could improve lipid synthesis, especially PUFAs synthesis. It showed that triclosan had the potential as a stimulator for improving lipid and PUFAs production in *S. limacinum.* This work provided a new vision for PUFAs synthesis at the molecular level and a future direction to reach a higher PUFAs production and total lipids.

## Figures and Tables

**Figure 1 microorganisms-08-00300-f001:**
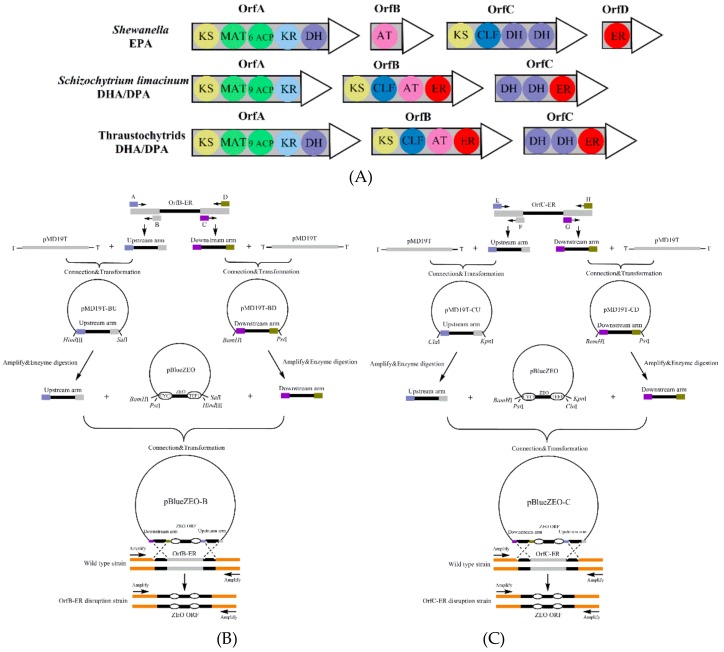
(**A**) Examples of polyunsaturated fatty acids (PUFAs) synthase organization in various representative organisms. KS: β-ketoacyl synthase; MAT: malonyl-CoA:ACP transacylase; ACP: acyl-carrier protein; KR: β-ketoreductase; DH: dehydratase; CLF: chain length factor; AT: acyl transferase; ER: enoyl-reductase; DH: dehydratase. (**B**) Construction of *OrfB-ER* gene konckout vector; (**C**) construction of *OrfC-ER* gene konckout vector.

**Figure 2 microorganisms-08-00300-f002:**
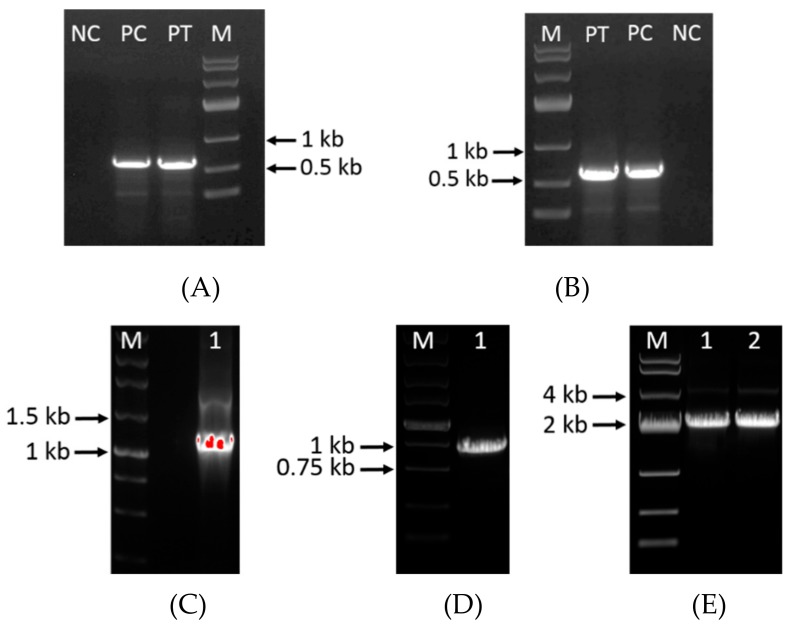
(**A**) Genomic PCR of partial zeocin expression cassette in the *OrfB-ER* disruption strain, (**B**) Genomic PCR of partial zeocin expression cassette in the *OrfC-ER* disruption strain. In the disruption strains, it should have an expected PCR product: Partial zeocin expression cassette (649 bp). PT: Positive transformants, PC: Positive control, NC: Negative control, M: Marker. (**C**) Genomic PCR of *OrfB-ER* gene (1334 bp) in the wild-type strain. (**D**) Genomic PCR of *OrfC-ER* gene (911 bp) in the wild-type strain. (**E**) Genomic PCR of upstream and downstream of *OrfB-ER* gene and *OrfC-ER* gene containing zeocin resistance cassette in the *OrfB-ER* and *OrfC-ER* disrupted strain, respectively. 1: Fragments of *OrfB-ER* and zeocin resistance cassette (1997 bp); 2: Fragments of *OrfC-ER* and zeocin resistance cassette strain (1972 bp).

**Figure 3 microorganisms-08-00300-f003:**
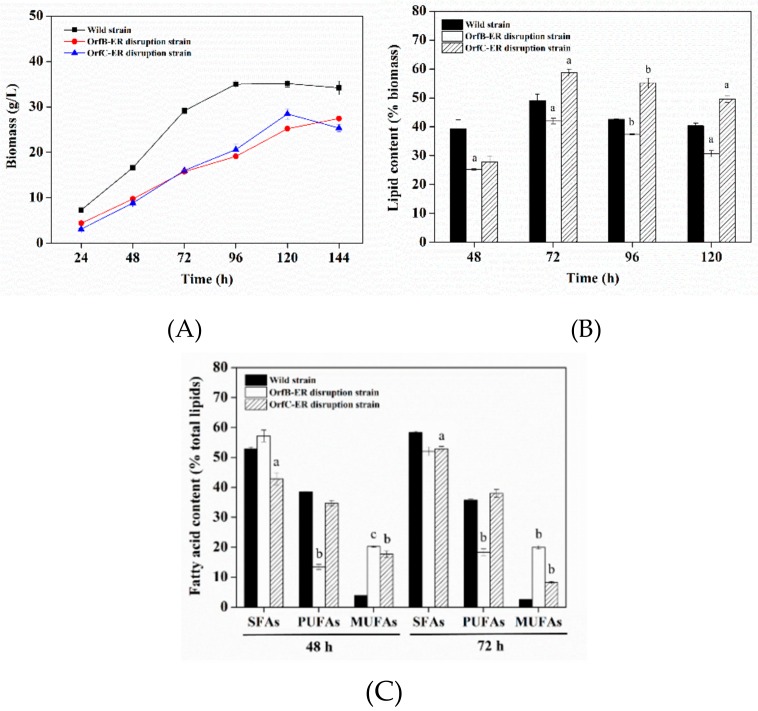
(**A**) Biomass, (**B**) lipid content of the biomass, and (**C**) fatty acid content of total lipid in the wild-type strain, *OrfB-ER* disruption strain, and *OrfC-ER* disruption strain. Significant difference between wild type strain and disruption strain is indicated at *p* < 0.05 (a), *p* < 0.01 (b), or *p* < 0.001 (c) level. All data are expressed as mean ± S.D. of three independent experiments.

**Figure 4 microorganisms-08-00300-f004:**
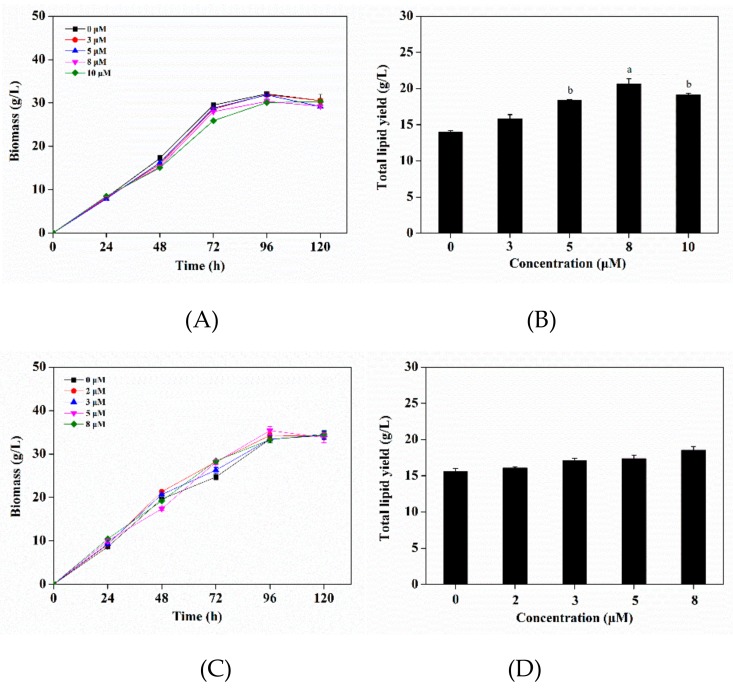
Effect of different concentration of triclosan added at 24 h on (**A**) biomass, (**B**) total lipid yield, and at 48 h on (**C**) biomass, (**D**) total lipid yield. Significant difference between control and triclosan treatment group is indicated at *p* < 0.05 (a), *p* < 0.01 (b) or *p* < 0.001 (c) level. All data are expressed as mean ± S.D. of three independent experiments.

**Figure 5 microorganisms-08-00300-f005:**
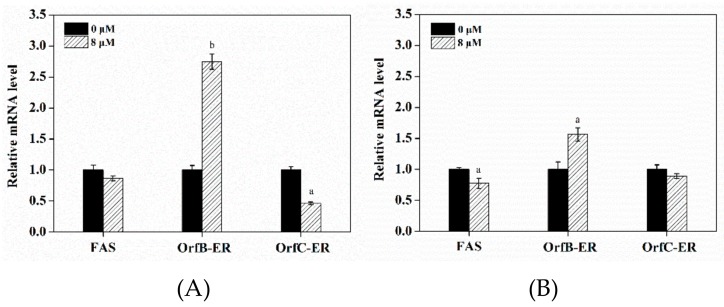
The gene expression profile of the fatty acid synthesis (*FAS*), *OrfB-ER* and *OrfC-ER* at (**A**) 48 h and (**B**) 72 h by RT-PCR from the control group and 8 μM triclosan treatment group. *FAS* means fatty acid synthase, *OrfB-ER* means enyolreductase located on *OrfB* subunit, *OrfC-ER* means enyolreductase located on *OrfC* subunit. Significant difference between the control and triclosan treatment group is indicated at *p* < 0.05 (a), *p* < 0.01 (b), or *p* < 0.001 (c) level. All data are expressed as mean ± S.D. of three independent experiments.

**Figure 6 microorganisms-08-00300-f006:**
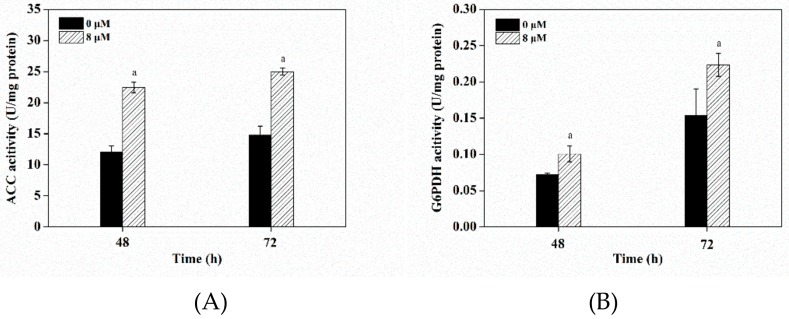
(**A**) Acetyl-CoA carboxylase (ACC) enzyme activities and (**B**) Glucose-6-phosphate dehydrogenase (G6PDH) enzyme activities in the control and 8 μM triclosan treatment. Significant difference between control and triclosan treatment group is indicated at *p* < 0.05 (a), *p* < 0.01 (b), or *p* < 0.001 (c) level. All data are expressed as mean ± S.D. of three independent experiments.

**Figure 7 microorganisms-08-00300-f007:**
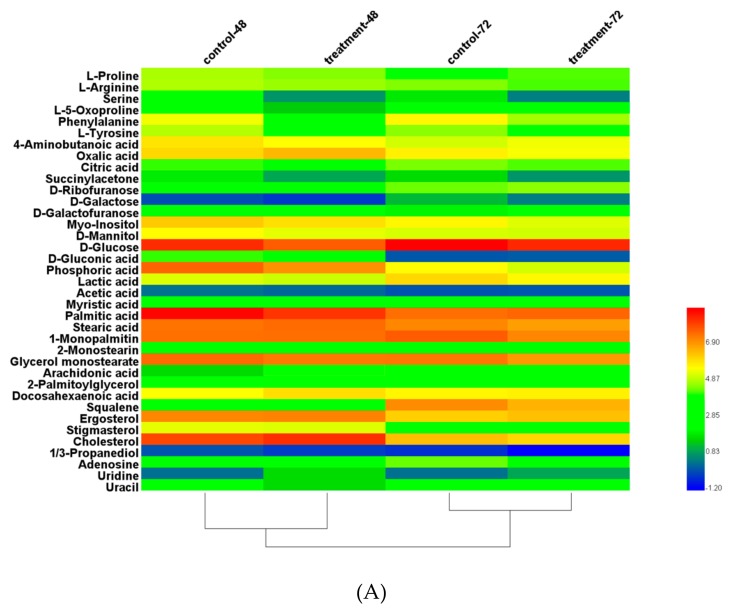
(**A**) Heat map for identified metabolites using log ratio. All data are expressed as the means of six replicates. (**B**) Significant changes of typical metabolites in main metabolic pathways induced by 8 μM triclosan in *S. limacinum*. Black squares and red squares represent the control group and 8 μM triclosan treatment, respectively. Significant difference between control and triclosan treatment group is indicated at *p* < 0.05 (a), *p* < 0.01 (b) or *p* < 0.001 (c) level. All data are expressed as mean ± S.D. of three independent experiments.

**Figure 8 microorganisms-08-00300-f008:**
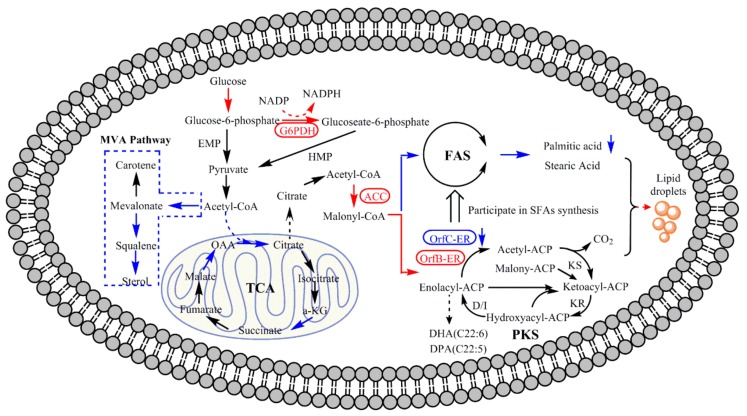
Functional display of *OrfB-ER* gene and *OrfC-ER* gene in fatty acid synthesis and central carbon metabolic regulation related to fatty acid biosynthesis when triclosan was added in *S. limacinum*. Red arrow means promoted; blue arrow means suppressed. Blue dashed box indicates that metabolic pathways are suppressed. Red oval box indicates enzyme activity increased. G6PDH: Glucose-6-phosphate dehydrogenase; ACC: Acetyl-CoA carboxylase; *OrfB-ER*: *ER* located on *OrfB* subunit; *OrfC-ER*: *ER* located on *OrfC* subunit; KS: Ketoacylsynthase; KR: β-ketoreductase; D/I: Dehydratase/Isomerase. OAA: oxaloacetic acid; a-KG: α-Ketoglutaric acid. MVA pathway: Mevalonate pathway; EMP: Glycolytic pathway. HMP: Hexose monophosphate pathway.

**Table 1 microorganisms-08-00300-t001:** Comparison of the total lipid and PUFAs yields between the control group and 8 μM triclosan treatment at 24 h. Significant difference between control and triclosan treatment group is indicated at *p* < 0.05 (a), *p* < 0.01 (b), or *p* < 0.001 (c) level. All data are expressed as mean ± S.D. of three independent experiments.

Lipid/PUFAs Yield	No Treatment	With Triclosan	Increase Ratio
Total lipids (g/L)	13.96 ± 0.26	20.61 ± 0.74^a^	47.6%
Total PUFAs (g/L)	4.89 ± 0.19	7.42 ± 0.06^b^	51.7%
EPA (mg/L)	53.75 ± 0.70	82.44 ± 1.09^a^	53.4%
DHA (mg/L)	4019.78 ± 154.26	6002.66 ± 9.27^a^	49.3%

**Table 2 microorganisms-08-00300-t002:** Lipid profiles (percentage) at 48 h and 72 h under control condition and 8 μM triclosan treatment at 24 h. Significant difference between control and triclosan treatment group is indicated at *p* < 0.05 (a), *p* < 0.01 (b), or *p* < 0.001 (c) level. All data are expressed as mean ± S.D. of three independent experiments.

Fatty Acid Composition	48 h		72 h	
	No Treatment	With Triclosan	No Treatment	With Triclosan
C14:0	3.68 ± 0.23	2.71 ± 0.17^a^	4.46 ± 0.12	3.45 ± 0.28^a^
C14:1	2.13 ± 0.14	3.20 ± 0.09^a^	1.42 ± 0.16	2.49 ± 0.06^a^
C16:0	49.19 ± 0.48	45.41 ± 0.38^a^	51.48 ± 0.39	49.34 ± 0.37^a^
C18:0	2.21 ± 0.25	1.84 ± 0.13^a^	1.73 ± 0.08	1.67 ± 0.07
EPA	0.26 ± 0.02	0.27 ± 0.05	0.21 ± 0.09	0.25 ± 0.02
DPA	5.86 ± 0.23	6.56 ± 0.36	5.16 ± 0.21	6.05 ± 0.23^a^
DHA	30.79 ± 0.56	33.68 ± 0.21^a^	30.26 ± 0.13	31.94 ± 0.48^a^
SFAs	55.53 ± 0.26	49.96 ± 0.36^a^	57.65 ± 0.13	55.33 ± 0.03^a^
PUFAs	36.95 ± 0.27	39.98 ± 0.58^a^	36.39 ± 0.14	38.08 ± 0.16^a^
PUFAs/SFAs	64.52 ± 0.13	76.20 ± 0.21^b^	60.30 ± 0.24	67.15 ± 0.18^a^

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
