# Peer review of "Functions of Enyolreductase (ER) Domains of PKS Cluster in Lipid Synthesis and Enhancement of PUFAs Accumulation in Schizochytrium limacinum SR21 Using Triclosan as a Regulator of ER"

_microorganisms, 2020, doi:10.3390/microorganisms8020300_

Round 1

Reviewer 1 Report

The manuscript "Functions of Enyolreductase (ER) Domains of PKS 2 Cluster in Lipid Synthesis and Enhancement of 3 PUFAs Accumulation in Schizochytrium limacinum 4 SR21 Using Triclosan as a Regulator of ER " is reporting on the role of two ER domains of the polyketide synthase cluster genes involved in fatty acid metabolism. 

The obtained data provide better understanding of PUFA metabolism in S. limacinum and will be helpful in the understanding of the regulatory mechanism of triclosan on the fatty acid metabolism.

In general, the manuscript contains poor English grammar. Especially the sections 2.2., 3.1., 3.2., 3.3., 3.4., and 3.5. Please, improve those parts significantly.

Comments to the Author:

Authors should carefully read the manuscript for typing errors, italize all latin names, use x g instead of rpm, use RT-qPCR instead of qRT-PCR, use wild type strain instead of wild strain, use concentration of triclosan instead of concentration triclosan, use compared to instead of compare with, avoid wording such as obvious throughout the manuscript and so on. Please check also for duplicated abbreviations in the manuscript.

Authors used strain of Schizochytrium limacinum for their study, but they use Schizochytrium sp. throughout the whole manuscript. Therefore, I suggest to use S. limacinum through the relevant part of the manuscript.

In the Material and methods section I miss details about the statistical analysis used to evaluate the data presented in the manuscript.

Providing of OrfB and OrfC sequence in the supplementary material will be beneficial.

Lines 94,95 - Primers for Figure 1B and 1C are listed in Table S1, however, they are not RT-qPCR primers as the title of Table S1 describes. Table S1 should be redesigned accordingly. In addition, the font of primers in Figure 1 is too small to read and the names of primers might be too long after enlargement. Therefore, I suggest to simplify their labelling as A, B, C, etc. or so in the Figure 1 and use the new labelling together with the present labelling in the Table S1.    

Line 94 - "The disrupted vector was constructed"..vector is not disrupted, therefore authors should change it to something like: vector used for disruption of genes...was constructed as.. Please, check also other places for such mistakes throughout the manuscript.

Line 97 - there should be no space between latin name of restriction enzymes and a number

Line 98 - the sentence should not start with "And"

Line 99 - "they were connected" replace with fragments were ligated

Line 105 - I miss the information which enzymes were used to linearize the vector

Line 106 - "2 kv" should be "2 kV"

Line 108 - "zeocin for growth", zeocin is not used for growth, please correct it

Line 116 - please, give details what FAMEs standards were used 

Line 132 - correct NaHCO3

Line 139 - give a short description how the sample was obtained for GC-MS analysis 

Line 153 - not only TEF1 promoter and CYC1 terminator were used to construct the plasmid for disruption, but also the zeocin ORF..

Lines 157-159 - I don`t agree with the authors statement. This sentence is not clear for me and based on the provided information and the size of PCR products on Figure 2 a) and b), it looks like the authors just checked the presence of the zeocin resistance cassette in the obtained transformants, but not its proper insertion in the S. limacinum genome. What kind of primers were used to check the insertion of cassettes? The checking primers should be listed in the Table S1. In addition, what means positive control in the Figure 2? 

If I am right and the authors just checked the presence of the zeocin cassette, I request authors to confirm the exact insertion of the disruption cassettes into OrfB-ER and OrfC-ER, respectively. 

Line 174 - last part of the Figure 1 legend should be rephrased

Lines 189-197 - should be rephrased. Authors should be careful with their statements - biomass production might not always describe the cell growth, because the cells can became just larger/heavier and therefore even the cells will not divide biomass will change. 

Line 225 and some other figure legends/table descriptions - what means "datum" and what do authors mean by "three independent experiments replicates"? Each experiment contained replicates or it was one experiment with three replicates?

Line 235 - "was to improve the"

Line 248 - "It would be studied"

Line 253 - "concentration triclosan" should be "concentration of triclosan"

Table 1 and 2 - please replace "Add triclosan" for "With triclosan" or "+ triclosan"

Line 268 - authors used improper English

Line 287 - I do not understand the statement "the triclosan weakened SFAs synthesis to enhance PUFAs synthesis". Please explain in more details.

Line 307 - "polyunsaturated fatty acids" "PUFAs"

Figure 7 - It is not clear enough which part of the figure is (a) and which part is (b)

Line 381 and 385- What authors mean by replicates in this case? How many experiments were performed? 

Figure 8 - Use detailed description of the figure. Also, please use abbreviations for KS, D/J, KR. How OrfC-ER participates in SFAs synthesis?

Figure S2 (c) please, check whether the sum of fatty acid content makes 100% in case of 3uM triclosan. Also correct “concentration triclosan” to “concentration of triclosan”

Reviewer 2 Report

The paper is engaging and well written in some parts. However, some details need to be clarified, and some mistakes corrected.

Introduction:

The very first sentence craves for more caution! The authors should carefully formulate the possible benefits and harms of PUFA regarding recent research literature. Some studies for consideration: 1) DOI:10.1016/j.plefa.2018.01.001 — about safety and tolerability of ω3 PUFA; 2) DOI:10.1016/j.jacl.2017.07.010 and DOI:10.3390/nu9080865 — ω3 PUFA in cardiovascular diseases prevention; 3) DOI:10.1080/1028415X.2017.1321813 — ω3 PUFA in Alzheimer's disease. Note that the effects are not always profound and obvious. “Malignant tumors” is a too broad term regarding the ω3 PUFA effects, in my opinion. In the abstract of Ref.2, there is a claim that DHA “has a positive effect… …on some cancers”. Note that the first author of Ref.2 has an obvious conflict of interest, since he represents the company, which produces DHA commercially.

The focus of the study on Schizochytrium sp. could be introduced in more convincing way. The authors should explain the benefits of using Schizochytrium sp. as PUFA producer, comparing with the recent results of different microalgae. Some recent studies for consideration: 1) DOI:10.1016/j.bcab.2019.01.017 and DOI:10.1186/s13068-018-1275-9 — reviews on lipid production by microalga; 2) DOI:10.1371/journal.pone.0224701, DOI: 10.1007/s10811-018-1471-9, and DOI:10.1002/biot.201900043 — recent reports on high PUFA production in microalgae; 3) DOI:10.1016/j.algal.2019.101619 — about the extraction and purification of EPA and DHA from microalgae.

The rest of the Introduction is fine, except for some awkward phrases. For example, “an important industrially alternative source” should be reformulated as “an industrially important source”.

Materials and methods:

How exactly were FAMEs prepared? Different derivatization protocols lead to the different results, especially when PFA are considered. What the standards were used for the column calibration? What was the detection and identification procedure? Please note that for the reliable identification, GC-MS is highly preferable. See also the comments on fatty acid composition results below.

Line 128 and further on: “at 10,000 rpm”. Regarding centrifugation, rpm values are meaningless unless the geometry of the rotor is reported. They should be replaced with rcf values (in g) reported.

Results and Discussion:

Table 2 shows surprisingly low variation in fatty acid profiles. It raises the following concerns: 1) How independent are the experiments reported in the Table 2? Were they conducted in parallel or consequently? 2) It would be more informative to provide not the percentage but yields of the fatty acids. Furthermore, reporting mean±SD of the percentage is somewhat misleading, because symmetric errors in yield of the individual components lead to asymmetric errors in percentage. 3) All detected fatty acids should be reported with equivalent chain length values (ECL). Preferably, the expanded table with data on all three independent experiments should be provided in supplementary.

Figure 7a could be improved. First, colorbar units should be explained in the caption. Second, the detected metabolites should be splitted to several heatmaps, e.g., low- and high-abundant metabolites. This way the effect of cultivation time and treatment would be clearer. Third, the metabolite labels should be of larger font size. Fourth, nomenclature should be consistent. For example, “Doconexent” should be indicated as DHA, and D- or L- prefixes should be all in capital letters. Last but not least, urea and ethanol are expected to elute early. Were these signals well separated from the solvent peak? Are authors sure in their correct identification? Why acetic acid is shown in Figure 7b but not in Figure 7a?

Round 2

Reviewer 1 Report

I thank authors for implementation of my suggestions to improve their manuscript.